# Association of Mediterranean Diet with Cardiovascular Risk Factors and with Metabolic Syndrome in Subjects with Long COVID: BioICOPER Study

**DOI:** 10.3390/nu17040656

**Published:** 2025-02-12

**Authors:** Nuria Suárez-Moreno, Leticia Gómez-Sánchez, Alicia Navarro-Caceres, Silvia Arroyo-Romero, Andrea Domínguez-Martín, Cristina Lugones-Sánchez, Olaya Tamayo-Morales, Susana González-Sánchez, Ana B. Castro-Rivero, Emiliano Rodríguez-Sánchez, Luis García-Ortiz, Elena Navarro-Matias, Manuel A. Gómez-Marcos

**Affiliations:** 1Primary Care Research Unit of Salamanca (APISAL), Salamanca Primary Care Management, Institute of Biomedical Research of Salamanca (IBSAL), 37005 Salamanca, Spain; nuria.suarez@usal.es (N.S.-M.); alicia.nav@usal.es (A.N.-C.); silvia_ar@usal.es (S.A.-R.); andreadm@usal.es (A.D.-M.); crislugsa@gmail.com (C.L.-S.); olayatm@usal.es (O.T.-M.); gongar04@gmail.com (S.G.-S.); anacasri@yahoo.es (A.B.C.-R.); emiliano@usal.es (E.R.-S.); lgarciao@usal.es (L.G.-O.); enavarro@saludcastillayleon.es (E.N.-M.); 2Castilla and León Health Service-SACYL, Regional Health Management, 37005 Salamanca, Spain; 3Emergency Service, University Hospital of La Paz P. of Castellana, 261, 28046 Madrid, Spain; letici.gomez@salud.madrid.org; 4Research Network on Chronicity, Primary Care and Health Promotion (RICAPPS), 37005 Salamanca, Spain; 5Department of Medicine, University of Salamanca, 28046 Salamanca, Spain; 6Department of Biomedical and Diagnostic Sciences, University of Salamanca, 37007 Salamanca, Spain

**Keywords:** mediterranean diet, cardiovascular risk factors, long COVID, metabolic syndrome

## Abstract

Background. Long COVID has been associated with increased cardiovascular risk and chronic low-grade inflammation, raising concerns about its long-term metabolic consequences. Given that the Mediterranean diet (MD) has shown beneficial effects on cardiovascular risk factors and inflammation in various populations, it is important to explore its potential impact on individuals with Long COVID. Therefore, the aim is to determine the association of the MD with cardiovascular risk factors (CVRF) and metabolic syndrome (MetS) in Caucasian subjects diagnosed with Long COVID. Methods. Cross-sectional study, 305 subjects diagnosed with Long COVID were included following the WHO criteria. Adherence to MD was evaluated with the MEDAS (Mediterranean Diet Adherence Screener) with 14 items used in Prevention with Mediterranean Diet study (PREDIMED study). The criteria considered to diagnose MetS were blood pressure, glycemia, triglycerides, HDL cholesterol, and waist circumference. Other CVRFs considered were tobacco consumption, total cholesterol, LDL cholesterol, body mass index, and baseline uric acid levels. The association between MD with CVRF and the number and components of MetS was analyzed using multiple regression models and multinomial regression. Results. The mean age was 52.75 ± 11.94 years (men 55.74 ± 12.22 and women 51.33 ± 11.57; *p* = 0.002), (68% women). The mean of the MEDAS questionnaire was 7.76 ± 2.37. The presented MetS were 23.6% (39.8% men and 15.9% women *p* < 0.001). In the multiple regression analysis, after adjusting for age and average time from acute COVID-19 infection to the date of inclusion in this study, the mean MD score showed a negative association with uric acid (β = −0.295; 95% CI: −0.496 to −0.093), BMI (β = −0.049; 95% CI: −0.096 to −0.002), the number of MetS components (β = −0.210; 95% CI: −0.410 to −0.010), and waist circumference (WC) (β = −0.021; 95% CI: −0.037 to −0.003) and a positive association with HDL cholesterol (β = −0.018; 95% CI: 0.001 to −0.037). Conclusions. The findings of this study suggest that higher Mediterranean diet scores are associated with lower levels of uric acid, fewer MetS components, smaller waist circumference, and higher HDL cholesterol levels in individuals with Long COVID.

## 1. Introduction

Dietary intake research is based on nutritional patterns [1], while the Mediterranean diet (MD) pattern is well-known and -researched. This diet includes a wide variety of foods such as extra virgin olive oil, legumes, cereals, dairy products, fish, nuts, fruits, vegetables, and red wine [2,3]. This type of food contains phytonutrients, polyphenols, and vitamins, which have anti-inflammatory and antioxidant effects [1,4,5]. Thus, several systematic reviews, meta-analyses, and studies have established the protective effects of MD on certain cardiovascular risk factors (CVRF), such are type 2 diabetes mellitus and obesity [6,7,8,9], dyslipidemia, and blood pressure [10,11]. These CVRFs increase the risk of cardiovascular disease and overall mortality, while MD components decrease morbidity and mortality [12,13]. Furthermore, MD has been shown to improve the immune system [14], as well as reduce the proinflammatory and prooxidative status in adults [15]. There are some studies which compare the benefits of this dietary pattern with others. Thus, the MD has been shown to outperform Western and standard low-fat diets, likely due to its emphasis on healthy fats, whole grains, and plant-based foods [16].

Metabolic syndrome (MetS) has become a major health problem [17]. Its prevalence is on the rise as a result of the increase in fast food consumption (which is high in calories and low in fiber) accompanied by scarce physical activity due to the increased use of mechanical transport and more sitting time in daily activities [18]. MetS doubles the risk of cardiovascular disease and increases the risk of all-cause mortality by 1.5 times [19,20,21]. CVRF, the components that define MetS (blood pressure, central obesity, dyslipidemia, and impaired glucose metabolism), and uric acid values increase the morbidity and mortality of COVID-19 in the acute phase [22,23]. Indeed, MetS increases COVID-19 mortality threefold in the acute phase [23], and SARS-CoV-2 infection increases insulin resistance in the acute phase [24]. Obesity activates the immune system, increasing inflammation and severity caused by the SARS-CoV-2 virus [25,26,27]. High blood pressure increases morbidity and mortality in the acute phase of COVID-19 infection [28]. Dyslipidemia facilitates the entry of SARS-CoV-2 into the cell, and low HDL cholesterol levels and increased triglycerides are associated with the severity of the disease [29,30,31,32]. Similarly, there are CVRF and MetS components that increase progression to Long COVID, such as smoking [33], dyslipidemia [34], arterial hypertension [34,35], diabetes mellitus [36], and obesity [34,37] and increased insulin resistance [38]. Therefore, in this phase of the pandemic, where subjects with Long COVID predominate, the relationship with MetS and other CVRFs is receiving greater attention [23,39,40]. This is how it is known that MetS influences the pathophysiological factors associated with Long COVID [23]. There are multiple pathophysiological processes in Long COVID, including viral factors (such as greater virus persistence, reactivation, and increased bacteriophagic activity); host-related factors (such as chronic inflammation, metabolic, endocrine, immune, and autoimmune alterations); and other effects such as different tissue damage from acute infection, tissue hypoxia, and alterations in the autonomic nervous system. All these mechanisms cause long-term endothelial inflammation and fibrin microclusters that affect the different organs of the human body [41]. We know that Long COVID and MetS interact with each other. This is shown by a recent study that summarizes the latest findings on the association between Long COVID and MetS. MetS influences different pathophysiological mechanisms associated with Long COVID. These mechanisms include reactions of the virus with pancreatic hormones, activation of the renin–angiotensin–aldosterone system, and increased oxidative stress causing a persistent inflammatory situation of the immune system. These pathological alterations contribute to the persistence of long-term symptoms, such as fatigue, dyspnea and other clinical manifestations [23,42]. All of the above demonstrates the existence of evidence that CVRF and MetS increase the progression of subjects infected with COVID-19 to Long COVID. However, the association of MD in these subjects with CVRF and MetS is not clear. Given the lack of information on the relationship between adherence to the Mediterranean diet and cardiovascular risk factors, as well as MetS and its components in people diagnosed with Long COVID, studies analyzing this relationship in this group of patients are needed. In this study, we therefore propose to analyze the association of the Mediterranean diet (MD) with CVRF and MetS in Caucasian subjects diagnosed with Long COVID.

## 2. Materials and Methods

### 2.1. Design

The main objectives of the study, biological determinants of Persistent COVID (BioICOPER), are to analyze the determinants, biopsychosocial, vascular structure and function, vascular aging, endothelial biomarkers, persistence of SARS-CoV-2 RNA and immune response in included subjects diagnosed with Persistent COVID following the criteria established by the WHO [41] and the differences by sex. This is a cross-sectional study in which, among other variables, MD and CVRF are analyzed, as described in the BioICOPER study protocol [43]. Persistent COVID Clinical Trial study registration: ClinicalTrials. NCT05819840 in April 2023.

### 2.2. Study Population

By consecutive sampling, 305 individuals meeting the criteria of the WHO’s clinical definition of Long COVID [44] were selected. Subjects are considered to have Long COVID when they have a history of SARS-CoV-2 infection, it is 3 months after onset, and they have different symptoms with a duration ≥2 months that cannot be explained by alternative diagnosis. Symptoms may appear as a new onset after initial recovery from an acute episode of COVID-19 or persist from the initial illness. In addition, these symptoms may fluctuate or recur over time [44]. The following were exclusion criteria: being unable to travel to health centers, having a terminal condition, a history of cardiovascular diseases such as heart disease, brain disease, or a glomerular filtration rate below 30%. Subjects were recruited from existing primary care clinic registries and the Long COVID clinic in the Salamanca health area’s internal medicine service. More details about the methodology of the BioICOPER study can be found in the published study protocol [43]. Sample size estimation: accepting an alpha risk of 0.05 and a statistical power greater than 0.8 in a two-tailed test, with a common standard deviation of 1.3, 84 subjects per group are needed to detect a statistically significant minimum difference of 0.7 in the number of MetS factors (Mean: 1.53 ± 1.33) among the pairs of the three groups (based on Mediterranean diet tertiles). Therefore, 305 subjects would be sufficient to test the study hypothesis. 

### 2.3. Variables and Measuring Instruments

All tests (questionnaires, clinical history data, physical examination, and laboratory analysis) were carried out at the Primary Care Research Unit in Salamanca within a maximum period of 10 days. The four researchers involved had previously received training following a standardized protocol, and data quality was revised by an independent researcher.

#### 2.3.1. Definition of Long COVID

Many terms have been used to describe this new entity; the most accepted have been Persistent COVID or Long COVID [37,45,46,47]. Currently, the most widely accepted definition is the one agreed upon by the WHO, which considers Long COVID to be the post-COVID condition that occurs in individuals with a history of probable or confirmed SARS-CoV-2 infection. This condition generally continues three months after the onset of COVID-19, with symptoms that last at least two months and cannot be explained by an alternative diagnosis. The most common symptoms are fatigue, shortness of breath, and cognitive dysfunction, but other symptoms that often affect quality of life can also occur. The symptoms may reappear after recovery from the acute process or may be continuous from the time of the infection. All symptoms can fluctuate, and relapses may occur over time [44].

#### 2.3.2. Mediterranean Diet

MD adherence was measured with the 14-item MEDAS. This questionnaire has been validated in Spain and was used in the PREDIMED study [48]. The questionnaire is composed of 12 questions on the frequency of food consumption and two items on dietary habits of the Spanish population. Each question was scored with 0 or 1 point, with 1 point for daily consumption of each of the following: (1) using olive oil as the principal fat for cooking, (2) ≥4 tablespoons of olive oil, (3) ≥2 servings of vegetables, (4) ≥3 servings of fruit, (5) <1 serving of red or meat processed meat, (6) <1 serving of animal fat, (7) <1 glass of carbonated or sugary drinks, and (8) eating more white meat than red meat. This is in addition to the weekly consumption of (1) ≥7 small glasses of wine, (2) ≥3 servings of legumes, (3) ≥3 servings of fish, (4) ≥3 servings of nuts, (5) ≥2 servings of sofrito, and (6) <2 servings of sweets or pastries per week. The total score ranges from 0 to 14 points. MD adherence is estimated with a score of 8 or more points (median value) [48].

#### 2.3.3. Metabolic Syndrome

To diagnose MetS, we followed the criteria established in the consensus of the Joint Scientific Declaration of the National Cholesterol Education Program III [49]. Table 1 shows the criteria, with a patient considered to have MetS when meeting > 2 of the following criteria [49].

#### 2.3.4. Diagnosis of Cardiovascular Risk Factors

Height was measured as the average of 2 readings on a calibrated wall-mounted height rod, with the patient standing upright, barefoot. The nearest value in cm was recorded. Body weight was taken twice with an approved and calibrated (accuracy ± 0.1 kg) electronic scale, with the patient barefoot. The body mass index (BMI) was calculated as weight (kg)/height (m^2^). Waist circumference (WC) was assessed with a tape measure above the iliac crest at the end of expiration and with the patient standing and unclothed. Three blood pressure measurements were taken with a validated Omron M10-IT sphygmomanometer (Omron Healthcare, Kyoto, Japan), with the average of the last 2 readings recorded. Measurements were taken on the participant’s dominant arm in a sitting position after at least 5 min of rest, using an appropriately sized cuff, determined by measuring the circumference of the upper arm, following the recommendations of the European Society of Hypertension (ESH) [50]. Blood samples were collected between 08.00 and 09.00 h after fasting overnight and without smoking or consumption of alcohol or caffeine in the previous 12 h. Plasma glucose, total cholesterol, high-density lipoprotein (HDL) cholesterol, and triglycerides were determined using automated standardized enzyme assays (Atellica™ Solution device. SIEMENS S.A—Erlangen, Germany—ISO 15189 [51]). Low-density lipoprotein (LDL) cholesterol was determined through the Friedewald formula [52]. Patients were considered to have hypertension if they were taking antihypertensive medications or had BP ≥ 140/90 mmHg. Type 2 diabetes mellitus was diagnosed if patients were taking hypoglycemic medications or had fasting plasma glucose levels ≥ 126 mg/dL or HbA1c ≥ 6.5%. Dyslipidemia was diagnosed if patients were taking lipid-lowering drugs or had fasting total cholesterol ≥ 240 mg/dL, low-density lipoprotein cholesterol (LDL-c) ≥ 160 mg/dL, high-density lipoprotein cholesterol (HDL-c) < 40 mg/dL in men and <50 mg/dL in women, or triglycerides ≥ 150 mg/dL. Obesity was diagnosed if they had a BMI value ≥ 30 kg/m^2^ [53].

### 2.4. Ethics Committee and Informed Consent

This project was approved by the Drug Research Ethics Committee of Salamanca on 27 June 2022 (CEIm reference code Ref. 2022/06). The participants in this work signed the informed consent form after being informed about the study and agreeing to participate. During the study implementation, the Declaration of Helsinki standards were followed [54].

### 2.5. Analysis

First, we checked whether the variables followed a normal distribution using the Kolmogorov–Smirnov test. To compare the means of independent samples between two categories, we used the Student’s *t*-test or the Mann–Whitney U test, depending on whether the distribution of the variables was normal or not. The comparison of two proportions was made using the Chi-square test. For comparisons of means of independent samples with more than two categories, we used either ANOVA or the Kruskal–Wallis H test, depending on whether the distribution of the variables was normal or not. Post hoc tests were performed using the DMS test. The correlation between the MD score and the CVRF, the number of MetS components, and their individual components was assessed using Spearman’s rho coefficient. The association between the MD score and the CVRF, the number of MetS components, and their individual components were analyzed using 13 multiple regression models. The MD score was used as the independent variable, while the CVRF, the number of MetS components, and the individual components were the dependent variables. Age (in years) and coded sex (0 = male, 1 = female) were used as adjustment variables, along with the average time from the acute COVID-19 infection to the inclusion in this study. The same models were used in a multinomial regression analysis, where the Mediterranean diet score was divided into tertiles, coded as tertile 1 = 0 (used as reference), tertile 2 = 1, and tertile 3 = 2. The same dependent and adjustment variables as in the multiple regression analysis were used. The SPSS Statistics program, version 28.0 (IBM Corp., Armonk, NY, USA), was used, with a *p*-value < 0.05 as the threshold for statistical significance.

## 3. Results

### 3.1. General Characteristics of Participants

Table 2 summarizes the variables analyzed in this study for the total sample and the differences by sex. The mean Mediterranean diet score was 7.76 ± 2.37 (7.69 ± 2.22 in men and 7.79 ± 2.44 in women). MetS was found in 23.6% (39.8% in men vs. 15.9% in women; *p* < 0.001). The following MetS components were more prevalent in men than in women: blood pressure (72.4% vs. 35.9%), plasma glucose (32.7% vs. 10.2%), and triglycerides ≥ 150 mg/dL (32.7% vs. 10.2%). The percentage of patients with hypertension, dyslipidemia, obesity, and type 2 diabetes mellitus was also higher in men. The average time from acute COVID-19 infection to the date of testing at the Research Unit was 38.66 ± 9.96 months.

Figure 1 represents the number and percentage of subjects by MD adherence and whether or not they were diagnosed with MetS. The highest percentage (48.2%) was found in the group of patients with adherence to MD and without MetS.

Figure 2 represents the percentage of patients according to the number of MetS criteria, overall and by sex. The highest percentage of women (33.2%) corresponds to those who do not meet any MetS criteria. The highest percentage of men (25.8%) corresponds to those who meet 4 MetS criteria.

Figure 3 shows the percentage of subjects with MetS and its components by tertiles of MD score overall. Appendix A display the same information by sex. Overall, the percentage of subjects with MetS and its components, except for hypertension, decreases as the MD score increases. However, this decrease is statistically significant only for the percentage of MetS (*p* = 0.050) and c-HDL (*p* = 0.015).

### 3.2. MD, Vascular Risk Factors and Metabolic Syndrome

Table 3 shows the characteristics of the subjects stratified into tertiles based on Mediterranean diet adherence score (Tertile 1 representing lower adherence and Tertile 3 representing the highest adherence). Subjects in Tertile 1 have higher uric acid values and lower c-HDL values compared to subjects in Tertile 2 and Tertile 3.

### 3.3. Correlation Coefficient Between the Mediterranean Diet Score with Cardiovascular Risk Factors and MetS Components

The Mediterranean diet score showed a positive correlation with clinical pulse pressure (PP) and HDL-c and a negative correlation with uric acid and waist circumference (WC) overall. In men, the Mediterranean diet score showed a negative correlation with WC, while in women, it showed a positive correlation with PP and HDL-c. In the partial correlation analysis adjusted for age, the Mediterranean diet score showed a positive correlation with HDL-c and a negative correlation with uric acid, BMI, and WC overall. In men, the Mediterranean diet score showed a negative correlation with both uric acid and WC. However, no correlation was found in women after adjusting for age Table 4.

### 3.4. Association Between Adherence to the Mediterranean Diet with Vascular Risk Factors and MetS and Its Components

In the multiple regression analysis, after adjusting for age and time since acute COVID-19 infection until inclusion in the study, the mean MD score showed a negative association with uric acid (β = −0.295; 95% CI: −0.496 to −0.093), BMI (β = −0.049; 95% CI: −0.096 to −0.002), number of MetS components (β = −0.210; 95% CI: −0.410 to −0.010), and WC (β = −0.021; 95% CI: −0.037 to −0.003) and a positive association with HDL-c (β = −0.018; 95% CI: 0.001 to −0.037) (Table 5).

Table 6 shows the association between the MD adherence with the CVRF and MetS components, using the first tertile of MD adherence as reference. 

## 4. Discussion

This is the first study in a Spanish population to analyze the association of MD with CVRF and the components of MetS in subjects diagnosed with Long COVID. Subjects in Tertile 1 had higher values of uric acid and lower levels of HDL-c compared to those in Tertile 2 and Tertile 3. The MD score showed a negative association with uric acid, BMI, the number of MetS components, and WC and a positive association with HDL-C.

In accordance with published data, the prevalence of Long COVID is higher in women than in men [54]. In contrast to the results of previous studies, which show that the mean MD score was higher in women than in men in different populations [53,54,55,56], including the Spanish population data [57], this study found no differences between sexes. This discrepancy may be explained by the small sample size and the imbalance between men and women. The proportion of people with Long COVID diagnosed with MetS was 27%, with a higher percentage in men. This proportion of subjects diagnosed with MetS was lower than data published on Hispanics aged over 60 years in the USA (57%) [55], in China 39% [56], or in Iran 37% [57] and similar to those published in Spain in the ENRICA study [58]. In the latter, however, contrary to the results of the present study, the proportion between sexes was different, with the percentage of subjects with MetS being higher in women (39% men and 44% women) [58]. Moreover, the percentage of MetS components found was higher in men than in women, results that are inconsistent with those published in other general population studies [58,59,60]. These discrepancies with previous studies are likely due to the higher percentage of women in our sample (68%) and the level of education (50% had university studies), in addition to coming from different geographic areas, different ages, the degree of obesity, and the definition applied for diagnosing MetS. The results are, therefore, not directly comparable. For all these reasons, prospective studies are needed that specifically analyze the effect of adherence to MD on CVRF and MetS in subjects with Long COVID. The difference between the groups with and without adherence to the MD and with and without MetS did not reach statistical significance (*p* = 0.077). However, the percentage of subjects in the four groups varied. Specifically, 12.5% of subjects had both MD adherence and MetS, while 28.2% of subjects had neither MD adherence nor MetS. We consider these differences, though not statistically significant, yet clinically relevant.

Our study found a negative association between MD and uric acid in subjects diagnosed with Long COVID, a link that has been confirmed in the general population [61]. Similarly, there are studies which have shown that high levels of uric acid are associated with adverse outcomes in subjects with COVID-19 in the acute phase of the disease [22,62]. For all these reasons, trials such as ALL-VASCOR are being proposed to test treatments with drugs to reduce uric acid levels, which will assess the incidence of Long COVID [63].

This study showed that MD was associated with the number of MetS components in subjects diagnosed with Long COVID (*p* = 0.039). This association has previously been found in the elderly Spanish population [64] and in the subjects included in the PREDIMED-PLUS study [65]. However, these results have not been consistent across all studies, as shown by results published in adults in Luxembourg [66] and in the study conducted by Hassani et al. [67], where MDA was not associated with MetS. Nevertheless, several studies that have implemented different interventions based on improving Mediterranean lifestyles have shown the beneficial effects of MD on MetS [68,69], although it should be noted that these studies did not estimate the effect of each of the components of the MD individually, but rather the overall, and possibly synergistic, effect of several related behaviors of Mediterranean culture. These interactions are probably conditioned by the maintenance of a chronic inflammatory state that causes a worse response of the immune system and intensifies not only the severity of the acute phase of COVID-19 but also contributes to its progression into Long COVID, which could, therefore, benefit from MD due to its anti-inflammatory and antioxidant properties [23]. Thus, more studies are needed to analyze the effect that each of the components of MD has on MetS.

In this work, MD showed an association with both general obesity (assessed by BMI) and abdominal obesity. These data are consistent with those published on the Spanish general population [70] or in a recently published meta-analysis [71]. To date, we have not found any research that analyzed this association in subjects diagnosed with Long COVID. However, there is evidence indicating a connection between obesity and weakened immune responses, which could facilitate the prolongation of the SARS-CoV-2 infection, increase its severity, and prolong the persistence of symptoms over time. This is related to chronic inflammation in different areas of the body, particularly in individuals with obesity, potentially impairing the immune system [23,27,70]. Such inflammation may be reduced with good MD adherence [23].

This work did not find an association between MD and glycemia, a result in line with Bakaloudi, DR et al. [71] but differing from data published on the Spanish population [70]. These differences can probably be explained by the prevalence of diabetes, the number of subjects, the sex ratio, and the age of the subjects included. There are studies suggesting that SARS-CoV-2 interacts with beta cells in the pancreas, which causes damage and impairment in insulin production while increasing insulin resistance. This process may favor the evolution of the symptoms of COVID-19, as well as the appearance of Long COVID, above all in people with pre-existing insulin resistance [72], thereby establishing a two-way association [73] and increasing severity due to underlying endothelial inflammation [74].

Similarly, we did not find an association between MD and blood pressure, a result consistent with previous studies [70,71]. Furthermore, high blood pressure can increase the long-term effects of COVID-19 [75]. This suggests that high blood pressure may increase the duration of Long COVID symptoms [76].

Finally, the MD adherence score is associated with higher HDL cholesterol levels but not with triglyceride levels in subjects diagnosed with Long COVID. These results differ from those found in various studies that have also reported an association with triglyceride levels [70,71]. Furthermore, unfavorable profiles (lower HDL cholesterol and higher triglycerides) are linked to the severity of the acute phase of the infection, while high-density lipoproteins (HDL) may benefit the control of COVID-19 infection, as well as its evolution to chronicity due to the beneficial effects, contrary to what occurs in subjects with high triglyceride levels [32,77,78].

In the Spanish population, studies with large population groups have analyzed the relationship between the Mediterranean diet and aging as well as different components of MetS, with similar results [52,71,79]. These results are also observed in other populations [80].

In summary, the MD can improve some components of MetS, although its effects seem to vary depending on the component evaluated. While our understanding of the role MD plays in Long COVID is expanding, further research is needed, particularly in patients with MetS. In these individuals, cardiovascular risk factors and the components of MetS must be controlled to prevent worsening and decrease the severity of Long COVID. This is essential because the bidirectional adverse effects of MetS and Long COVID seem evident, and there are a large number of people affected by MetS who experience the consequences of Long COVID, which requires an approach with multiple strategies to address the individual care of subjects with Long COVID. In this, the potential role of MD and its components should not be forgotten since its anti-inflammatory, antioxidant, and immune system effects can improve the health of these patients [23]. We must, therefore, ensure suitable nutrition for these subjects, increasing protein intake and correcting vitamin and micronutrient deficiencies. In addition, strict control of MetS components during acute COVID-19 will reduce the development of Long COVID and help control it [23].

Finally, it is important to highlight that the Mediterranean diet can influence the components of metabolic syndrome due to its high content of dietary fiber, omega-3 and omega-9 fatty acids, complex carbohydrates, antioxidants, minerals, vitamins, and bioactive substances such as polyphenols. These nutrients and bioactive substances can help combat obesity, dyslipidemia, hypertension, and diabetes mellitus. The mechanisms by which they act are generally related to oxidative stress, inflammation (the most common risk factors for metabolic syndrome), and gastrointestinal function [81].

Lastly, a recent review indicates that for the prevention and treatment of MetS, it is recommended to increase the daily intake of fiber-rich and low-glycemic index foods, as well as the consumption of fish and dairy products—especially yogurt and nuts—and to consume a wide variety of cereals, legumes, and unprocessed fruits, all of which are components of the MD [82]. This reaffirms the importance of recommending adherence to the MD as a healthy lifestyle in healthcare consultations.

### Limitations and Strengths

This research has some important limitations: (1) it is a cross-sectional study, which does not allow causality to be established; (2) the sample studied is not very large, with a large predominance of women, and the analysis between sexes must, therefore, be made with caution; (3) the use of a cut-off point for MD adherence is arbitrary; (4) the data on MD eating habits were gathered through questionnaires. That said, the study offers some strengths, such as the use of a definition of Long COVID with global consensus [44] and of measurements carried out following standardized protocols and with prior training of researchers, who took measurements with validated and calibrated devices.

## 5. Conclusions

The results of this study suggest that higher Mediterranean diet scores are associated with lower levels of uric acid, a lower mean number of MetS components, smaller waist circumference, and higher HDL cholesterol levels in subjects with Long COVID. Although these findings are descriptive, they highlight the potential role of the Mediterranean diet in decreasing cardiovascular risk in this population. However, to confirm these results, further prospective studies with a larger number of patients are needed.

## Figures and Tables

**Figure 1 nutrients-17-00656-f001:**
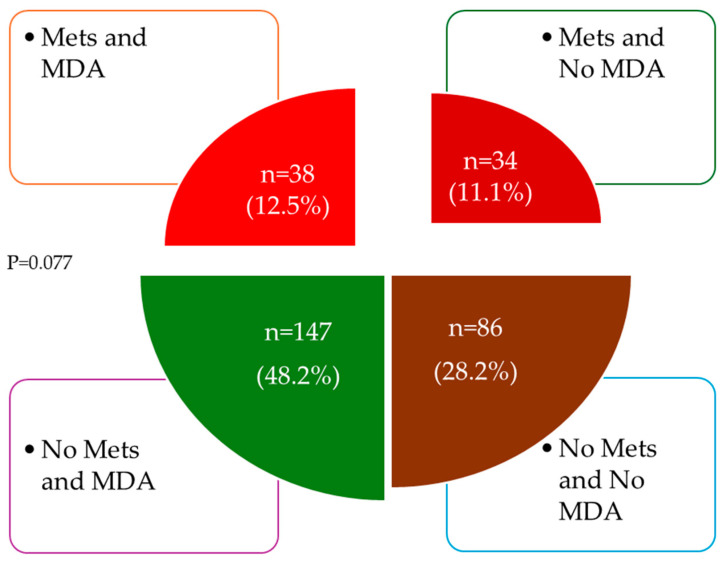
Percentage and number of patients according to MetS and/or MDA. MDA: Mediterranean diet adherence; MetS: metabolic syndrome. Good adherence was considered if the Mediterranean diet score was greater than 8.

**Figure 2 nutrients-17-00656-f002:**
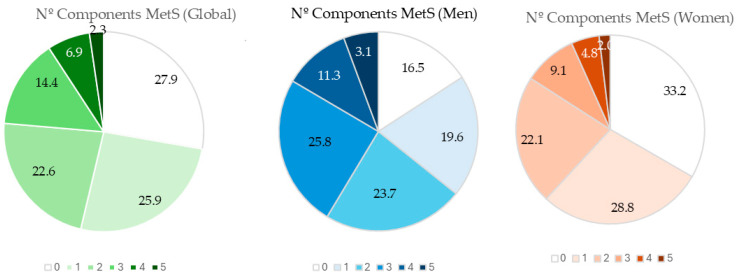
Percentage of subjects according to the number of components of metabolic syndrome. MetS: metabolic syndrome.

**Figure 3 nutrients-17-00656-f003:**
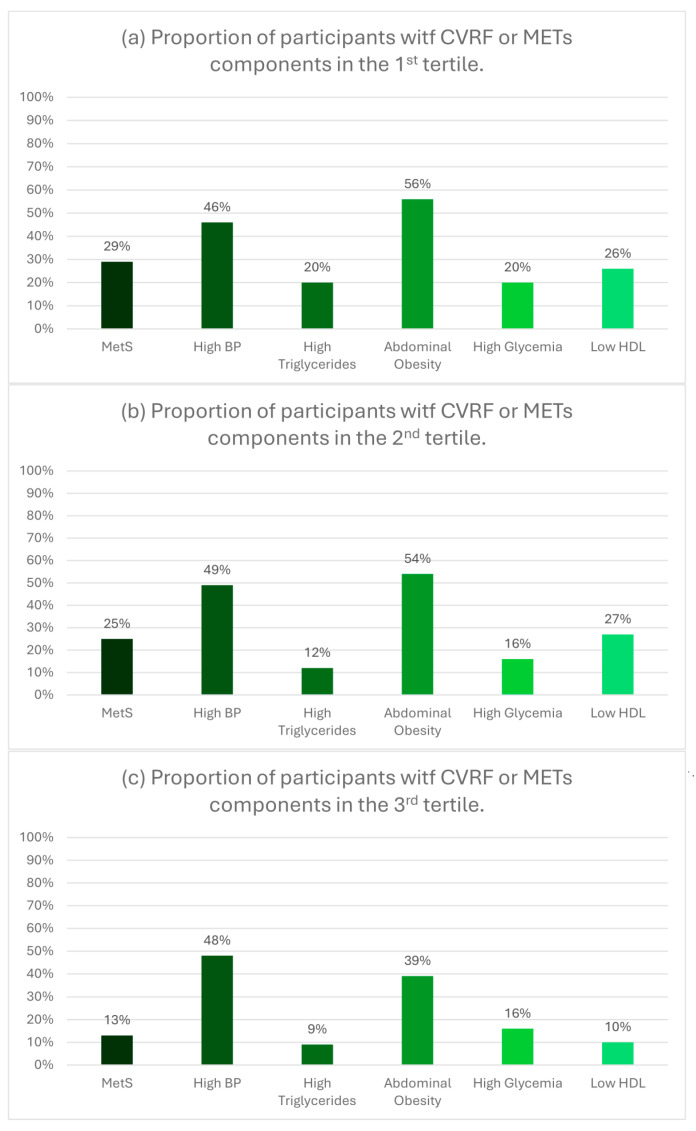
Percentage of MetS and its components across the tertiles of the MD score. MetS: metabolic syndrome. CVRF: cardiovascular risk factors. HDL: high-density lipoproteins.

**Table 1 nutrients-17-00656-t001:** MetS.

Blood Pressure	BP ≥ 130/85 mmHg or Treatment with Antihypertensive Drugs
Glycemia	FBG ≥ 100 mg/dL or treatment with hypoglycemic medication
Triglycerides	TGC ≥ 150 mg/dL or treatment with lipid-lowering medication
HDL cholesterol	HDL-C < 40 mg/dL in men or <50 mg/dL in women
Waist circumference	WC ≥ 88 cm in women or WC ≥ 102 cm in men

MetS: metabolic syndrome; BP: blood pressure; FBG: fasting plasma glucose; TGC: triglycerides; HDL-C: high-density lipoprotein cholesterol; WC: waist circumference. Adapted from the International Diabetes Federation Task Force on Epidemiology and Prevention; National Heart, Lung, and Blood Institute; American Heart Association; World Heart Federation; International Atherosclerosis Society; and International Association for the Study of Obesity [49].

**Table 2 nutrients-17-00656-t002:** General characteristics of the subjects included overall and by sex.

	Overall		Men		Women		
MD	Mean or n	SD or (%)	Mean or n	SD or (%)	Mean or n	SD or (%)	*p* Value
MD (total score)	7.8	2.4	7.7	2.2	7.8	2.4	0.438
Evolution time, months	38.7	9.6	38.7	9.6	38.7	9.6	0.990
Conventional risk factors							
Sex, n (%)	---	----	98	(32)	207	(68)	<0.001
Age (years)	52.8	11.9	55.7	12.2	51.3	11.6	0.002
No. cigarettes (per day)	6.0	10.2	9.1	12.4	4.5	8.5	<0.001
Smoker, n (%)	8	(8.3)	10	(5.1)	18	(6.1)	0.272
SBP (mmHg)	120.1	16.9	129.8	14.5	115.6	16.0	<0.001
DBP (mmHg)	76.9	11.1	82.3	11.0	74.3	10.2	<0.001
PP (mmHg)	43.3	10.3	47.5	10.9	41.3	9.3	<0.001
Uric Acid (mg/dL)	5.0	1.2	5.9	1.1	4.5	1.1	<0.001
Hypertension, n (%)	110	(36.2)	53	(54.1)	57	(27.7)	<0.001
Antihypertensive drugs, n (%)	79	(26.0)	34	(34.7)	45	(21.8)	0.012
Total cholesterol (mg/dL)	187.5	34.5	181.9	32.9	190.2	35.0	0.070
LDL cholesterol (mg/dL)	112.7	30.8	113.2	32.3	112.5	30.2	0.422
HDL cholesterol (mg/dL)	57.0	13.7	48.9	10.9	60.8	13.2	<0.001
Triglycerides (mg/dL)	102.1	50.5	116.9	54.4	95.1	47.0	<0.001
Dyslipidemia, n (%)	168	(56.0)	64	(66.0)	104	(51.2)	<0.001
Lipid–lowering drugs, n (%)	75	(24.8)	40	(41.2)	35	(17.1)	<0.001
FPG (mg/dL)	87.9	17.7	94.3	19.7	84.9	15.8	<0.001
Diabetes mellitus, n (%)	37	(12.2)	22	(22.4)	15	(7.3)	<0.001
Hypoglycemic drugs, n (%)	32	(10.5)	18	(18.4)	14	(6.8)	0.003
Weight (kg)	75.9	17.4	88.1	14.9	70.2	15.4	<0.001
Height (cm)	164.6	8.8	172.6	7.4	160.7	6.5	<0.001
BMI (kg/m^2^)	28.0	5.5	29.6	4.6	27.2	5.8	<0.001
WC (cm)	93.9	15.5	104.4	12.5	88.9	14.3	<0.001
Obesity, n (%)	98	(32.1)	44	(44.9)	54	(26.1)	<0.001
MetS and its components							
Number of MetS components	1.5	1.3	2.1	1.4	1.3	1.3	<0.001
MetS, n (%)	72	(23.6)	39	(39.8)	33	(15.9)	<0.001
BP ≥ 130/85 mmHg, n (%)	145	(47.7)	71	(72.4)	74	(35.9)	<0.001
FPG ≥ 100 mg/dL, n (%)	53	(17.5)	32	(32.7)	21	(10.2)	<0.001
TGC ≥ 150 mg/dL, n (%)	43	(14.1)	21	(21.4)	22	(10.7)	0.012
HDL-C < 40 mg/dL men, <50 mg/dL women, n (%)	70	(23.0)	23	(23.5)	47	(22.8)	0.832
WC ≥ 88 cm women, ≥102 cm men, n (%)	157	(51.5)	53	(54.1)	104	(50.2)	0.532

Values are means and standard deviations for continuous data and number and percentages for categorical data. MD: Mediterranean diet; SBP: systolic blood pressure; DBP: diastolic blood pressure; PP: pulse pressure; LDL: low-density lipoprotein; HDL: high-density lipoprotein; FPG: fasting plasma glucose; BMI: body mass index; WC: waist circumference; MetS: metabolic syndrome; TGC: triglycerides; No. cigarettes: number of cigarettes per day reported by participating smokers. *p* value: differences between men and women. Chi-square tests were used to compare percentages, and Student’s *t*-test or Mann–Whitney U test was used to compare continuous variables.

**Table 3 nutrients-17-00656-t003:** Values of vascular risk factors and MetS components across tertiles of Mediterranean diet.

	1st Tertile		2nd Tertile		3rd Tertile		
	Mean	SD	Mean	SD	Mean	SD	*p* Value
CVRF							
Cigarettes per day (n)	15.6	11.4	15.3	10.8	15.5	10.4	0.991
PP (mmHg)	42.4	11.1	43.0	9.6	44.9	10.8	0.300
Uric acid (mg/dL) ^a,b,c^	5.2	1.5	5.0	1.2	4.6	1.1	0.011
Total cholesterol (mg/dL)	190.4	38.0	187.2	31.9	184.5	36.5	0.586
LDL cholesterol (mg/dL)	114.4	33.3	113.4	30.2	109.7	30.6	0.635
BMI (kg/m^2^)	28.5	5.8	28.1	5.5	27.0	5.5	0.261
MetS							
MetS components (n) ^b^	1.7	1.4	1.6	1.4	1.3	1.2	0.149
SBP (mmHg)	119.7	16.5	120.2	16.6	120.2	18.2	0.976
DBP (mmHg)	77.4	10.7	77.2	11.6	75.3	10.3	0.450
FPG (mg/dL)	88.4	13.6	87.9	18.0	87.2	21.1	0.922
Triglycerides (mg/dL)	112.7	54.7	99.2	46.8	96.6	52.7	0.092
HDL cholesterol (mg/dL) ^a,b,c^	56.0	14.4	56.0	13.2	60.6	13.4	0.029
WC (cm) ^a^	96.0	16.5	94.0	15.3	90.9	14.5	0.139

CVRF: cardiovascular risk factors; PP: pulse pressure; LDL: low-density lipoprotein; BMI: body mass index; MetS: metabolic syndrome; SBP: systolic blood pressure; DBP: diastolic blood pressure; FPG: fasting plasma glucose; HDL: high-density lipoprotein; WC: waist circumference. No. cigarettes: number of cigarettes per day reported by participating smokers. *p* value: statistically significant differences between the three groups (*p* < 0.05). The ANOVA test was used for variables with a normal distribution, and the Kruskal–Wallis H test was applied for variables without normal distribution. Post hoc contrasts: ^a^ Between ‘1st tertile’ and ‘2nd tertile’. ^b^ Between ‘1st tertile’ and ‘3rd tertile’. ^c^ Between ‘2nd tertile’ and ‘3rd tertile’.

**Table 4 nutrients-17-00656-t004:** Correlation coefficient between MD score with vascular risk factors and number and components of MetS, overall, by sex, and age-adjusted.

MD	Cigarettes	MetS Number	SBP	DBP	PP	UA	Total-c	LDL	HDL	TGC	FPG	BMI	WC
Global	−0.002	−0.083	−0.057	−0.016	0.149 **	−0.117 *	0.014	0.008	0.143 *	−0.065	0.061	−0.099	−0.123 *
Men	−0.016	−0.135	−0.073	−0.076	0.075	−0.174	−0.065	−0.027	0.072	−0.179	−0.072	−0.158	−0.212 *
Women	0.059	−0.028	−0.029	0.016	0.202 **	−0.02	0.035	0.022	0.137 *	−0.015	−0.024	−0.057	−0.064
Partial correlation													
Global	0.007	−0.134 *	−0.018	−0.072	0.051	−0.161 *	−0.007	0.006	0.115 *	−0.069	−0.012	−0.133 *	−0.151 *
Men	−0.009	−0.111	−0.081	−0.116	0.015	−0.248 *	−0.024	0.008	0.050	−0.090	−0.003	−0.177	−0.213 *
Women	0.026	−0.126	−0.035	−0.029	0.094	−0.132	−0.054	−0.036	0.115	−0.053	−0.028	−0.102	−0.104

Correlation between the MD score with CVRF and number of MetS components. SBP: systolic blood pressure; DBP: diastolic blood pressure; PP: pulse pressure; UA: uric acid; Total-c: total cholesterol; LDL: low-density lipoprotein; BMI: body mass index; MetS: metabolic syndrome; FPG: fasting plasma glucose; HDL: high-density lipoprotein; TGC: triglycerides; FPG: fasting plasma glucose; WC: waist circumference; cigarettes: number of cigarettes per day reported by participating smokers. * *p* < 0.005. ** *p* < 0.001. The correlation with its components was performed using Spearman’s rho coefficient.

**Table 5 nutrients-17-00656-t005:** Association of the MD with CVRF and metabolic syndrome. Multiple regression analysis.

CVRF	β	(95%	CI)	*p*
No. Cigarettes (per day)	0.004	−0.035	0.042	0.854
PP (mmHg)	0.017	−0.010	0.044	0.215
Uric acid (mg/dL)	−0.295	−0.496	−0.093	0.004
Total cholesterol (mg/dL)	0.000	−0.008	0.007	0.905
LDL cholesterol (mg/dL)	0.000	−0.008	0.009	0.923
BMI (kg/m^2^)	−0.049	−0.096	−0.002	0.042
MetS	β	(IC	95%)	*p*
Number of MetS components	−0.210	−0.410	−0.010	0.039
SBP (mmHg)	−0.001	−0.018	0.016	0.892
DBP (mmHg)	−0.016	−0.041	0.008	0.187
FPG (mg/dL)	−0.001	−0.017	0.014	0.847
Triglycerides (mg/dL)	−0.003	−0.008	0.002	0.232
HDL cholesterol (mg/dL)	0.018	0.001	0.037	0.050
WC (cm)	−0.021	−0.037	−0.003	0.021

Multiple regression analysis using the number of MetS components, SBP, DBP, FPG, triglycerides, HDL cholesterol, and WC as dependent variables; the Mediterranean diet score as the independent variable; and age and time from the acute COVID-19 infection to the inclusion in the study as adjustment variables. MetS: metabolic syndrome; MD: Mediterranean diet; SBP, systolic blood pressure; DBP, diastolic blood pressure; HDL, high-density lipoprotein; FPG, fasting plasma glucose; WC, waist circumference. No. cigarettes: number of cigarettes per day reported by participating smokers.

**Table 6 nutrients-17-00656-t006:** Association between the Mediterranean diet and tertiles with CVRF and MetS. Multinomial regression analysis.

	1st Tertile	2nd Tertile			3rd Tertile		
CVRF		β	IC 95%		β	IC 95%	
No. cigarettes	Reference	1.013	0.970	1.057	1.014	0.960	1.071
PP	Reference	1.008	0.979	1.038	1.030	0.995	1.066
Uric acid	Reference	0.854	0.690	1.057	0.657	0.497	0.868
Total cholesterol	Reference	0.997	0.990	1.005	0.995	0.986	1.005
BMI	Reference	0.989	0.943	1.038	0.952	0.894	1.013
MetS							
MetS components	Reference	0.911	0.743	1.116	0.761	0.584	0.992
SBP	Reference	1.003	0.985	1.020	1.004	0.983	1.025
DBP	Reference	0.999	0.974	1.025	0.983	0.952	1.014
FPG	Reference	0.999	0.984	1.014	0.987	0.977	1.017
Triglycerides	Reference	0.995	0.990	1.000	0.954	0.987	1.001
HDL cholesterol	Reference	0.999	0.979	1.021	1.024	1.000	1.050
WC	Reference	0.995	0.974	1.024	0.994	0.956	1.000

Multinomial regression analysis using the Mediterranean diet score as the independent variable, divided into tertiles. Codified: Tertile 1 = 0 (used as reference), Tertile 2 = 1, and Tertile 3 = 2. Dependent variables: number of MetS components, SBP, DBP, FPG, triglycerides, HDL cholesterol, and WC. Adjustment variables: age and time from acute COVID-19 infection to study inclusion. MetS: metabolic syndrome; SBP, systolic blood pressure; DBP, diastolic blood pressure; HDL, high-density lipoprotein; FPG, fasting plasma glucose; WC, waist circumference. CVRF: cardiovascular risk factors; PP: pulse pressure; BMI: body mass index; MetS: metabolic syndrome; SBP: systolic blood pressure; DBP: diastolic blood pressure; FPG: fasting plasma glucose; HDL: high–density lipoprotein; WC: waist circumference. No. cigarettes: number of cigarettes per day reported by participating smokers.

## Data Availability

The data supporting the findings of this study are available on ZENODO under the DOI. 10.5281/zenodo.14282873.

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
