# Peer review of "Association of Mediterranean Diet with Cardiovascular Risk Factors and with Metabolic Syndrome in Subjects with Long COVID: BioICOPER Study"

_nutrients, 2025, doi:10.3390/nu17040656_

Round 1
Reviewer 1 Report (New Reviewer)
Comments and Suggestions for Authors
This study offers a compelling exploration of the relationship between the Mediterranean diet (MD), cardiovascular risk factors (CVRF), and metabolic syndrome (MetS) in individuals experiencing Long COVID. By thoroughly documenting population demographics and clinical variables and employing robust statistical analyses, the study underscores the potential significance of MD in managing these conditions. Nonetheless, several key points warrant further consideration:
1. The study population is predominantly female, which could restrict the generalizability of its findings. Would a more balanced representation of sexes or detailed subgroup analyses strengthen the validity of the conclusions?
2. As a cross-sectional study, it inherently limits the ability to draw causal inferences. Would adopting a prospective or longitudinal design better clarify the temporal relationship between MD adherence and CVRF or MetS in Long COVID patients?
3. While the study links MD adherence to favorable changes in HDL cholesterol, uric acid, BMI, and waist circumference, the biological mechanisms underlying these associations are not addressed. Could incorporating molecular biomarkers or inflammatory pathway analyses provide deeper insights?
4. The MEDAS questionnaire is a widely used tool for evaluating MD adherence, but its reliance on self-reported data introduces the possibility of bias. Would the inclusion of objective dietary biomarkers enhance the reliability of the findings?
5. Although the analysis adjusts for age and time since COVID-19 infection, other potential confounding variables, such as physical activity levels and socioeconomic factors, are not accounted for. Could incorporating these variables refine the observed associations?
6. While the study establishes associations between MD adherence and CVRF or MetS components, it does not address the feasibility of dietary interventions in Long COVID management. Could pilot studies testing MD-based interventions offer practical insights into its clinical utility?
7. The study’s findings are not compared extensively with other populations or dietary approaches. Would analyzing similar studies in different geographic or cultural settings, or comparing MD with alternative dietary patterns, broaden the applicability of the results?
Author Response
See attached file

Reviewer 2 Report (New Reviewer)
Comments and Suggestions for Authors
The manuscript accounts of a study of the the association of the Mediterranean diet) with cardiovascular risk factors and metabolic syndrome in Caucasian subjects diagnosed with Long COVID. Multiple regression analysis of the results showed a significant negative association between the mean Mediterranean diet score and with uric acid, number of metabolic syndrome components and waist circumference, and a positive association with HDL cholesterol.
The study, including 305 subjects, was properly designed, the definition of Long Covid was based on the WHO criteria, and properly elaborated statistically. The limitations of the study are discusses; the conclusions are scientifically sound.
Remarks:
Please provide a short description of methods of blood analysis. Do the reported values refer to blood plasma or serum? (it is obvious only for FPG)
Line 32: Please replace „y” by „and”
Line 38: please explain “WC” (which is explained later, in Table legend)
Table 2: Please explain “PP”; the acronyms should be explained on the first use
Table 2: “No. cigarettes, (per day)”, were only smokers considered, or is it an average for all participants?
Tables: please use “dL” instead of “dl”
Line 381: “B cells”, apparently beta cells
References: please use journal abbreviations with periods.
Author Response
See attached file

This manuscript is a resubmission of an earlier submission. The following is a list of the peer review reports and author responses from that submission.
Round 1
Reviewer 1 Report
Comments and Suggestions for Authors
This is a well-conducted study aimed to determine the relationship of the Mediterranean diet (MD) with cardiovascular risk factors and metabolic syndrome in Caucasian subjects diagnosed with Persistent COVID.
My comments:
1. In the text, I recommend changing relationship to association since you run multiple regression analyses.
2. Also in the title, the “Bioicoper study” seems a little lost. I recommend removing it from the title or a better detail in the methods section.
3. In the abstract section, please cite the possible confounder used as adjustment.
4. In the introduction, the relationship between metabolic syndrome and Covid in the acute phase of the disease is clear. but it is necessary to provide more detail on the pathophysiological relationship between metabolic syndrome and persistent covid to justify the objective of this study. I recommend that the authors summarize the information about the acute phase and focus on the changes in the persistent phase.
5. In the 2.5 item, does the data was subjected to the normality test (Shapiro- Wilk)? The authors only presented parametric analysis.
6. How long has the sample been diagnosed with persistent covid?
7. In the 3.1 item, the authors declares “Patients with good MD adherence comprised 60.7% of the sample”. But if the cutpoint to adherence according to MEDAS is 8, how the authors define a good adherence if the mean was 7.76±2.37? Please clarify.
8. Figure 1 shows the groups studied with no statistical difference. but considering the cutoff point of 8 for adherence to MD, how do the authors explain this figure?
9. In Table 3, I still believe that the dichotomic result as adherence and non-adherence is not adequate considering the MEDAS cutoff point, I suggest that the authors analyze the score results by tertile or quintile to identify possible differences between the variables.
10. In the discussion section, I suggest the authors include the clinical implications of their results for the health professionals. How these results can help in clinical and nutritional managing patients and improving Mets parameters?
Author Response
Reviewers' comments: Authors' Answer
Reviewer #1:

Reviewer 2 Report
Comments and Suggestions for Authors
Nuria Suárez-Moreno et al. documented the relationship of the Mediterranean diet (MD) with cardiovascular risk factors and with metabolic syndrome in subjects diagnosed with persistent COVID-19. The topic is very interesting. Although the experiments were done on over 300 patients, the group has a vast disproportion between males and females. The criteria for Mediterranean diet adherence are unclear, although the reference is provided. The conclusions were not drawn well and should be corrected.
I think the presented data does not meet the requirements in Nutrients. Below my comments.
Abstract
-Please explain what is PREDIMED study
-“MetS was defined based on the Joint Scientific Statement National Cholesterol Education Program III” – this statement is not necessary in the abstract, but it should be placed in the Method Section
-“Clinical Trial registration: ClinicalTrials.gov, identifier NCT05819840” - should also be placed in the Method Section.
-The conclusion have to be corrected.
Methods
-It would be helpful for the readers to briefly define symptoms of persistent COVID-19 in the introduction or method section, although you put a specific reference.
-Your statement below explains the clinical definition of persistent COVID-19, but it would be easier for the readers to explain this criterion briefly.
“By consecutive sampling, 305 individuals meeting the criteria of the WHO’s clinical definition of Persistent COVID, established by international Delphi consensus [41].”
-The crucial is how long the patients suffer from persistent COVIT-19. Please put this information in the table 2.
Results
-Please explain the p-value and appropriate statistical tests in the Tab. 2, 3 descriptions.
-An essential limitation of your study is the disproportion between men (98) and women (207). The mean adherence to MDA was similar in both groups, but MetS was more prevalent in men (39.8%) than in women (15,9%).
-Table 3 needs the number of patients which meets/do not meet MDA.
“Subjects with adherence to MD compared to nonadherent subjects had lower levels of uric acid (4.8 vs 5.2 mg/dl), triglycerides (98 vs 108 mg/dl), weight (74 vs 78 kg), waist circumference (92 vs 97 cm) and a lower percentage of subjects with obesity (28% vs 38%), with MetS (20% vs 28%) and with triglycerides ≥ 150mg/dl (11% vs 19%).”- you did not reach the scientific signiphicance in triglycerides and Met S.
- The multiple regression analysis has to be corrected. You did not reach statistical significance in the described parameters regarding the number of MetS components. A p-value is less than 0,05 only in uric acid and HDL. A beta value is minimal, and the effect is not meaningful, so this information has to be delivered in your conclusion. The size of the beta indicates the strength of the relationship. The beta values are small, which implies small changes. Even small positive or negative effects depend on statistical significance.
- Enrolled group of patients (female vs. male) initially varied in different parameters (smoking, blood pressure, uric acid, dyslipidemia, DM, obesity, and MetS components). When dividing them into two groups with/without Mediterranean diet adherence, those differences in the number of females and males are still present (the group with adherence: 55 males vs. 130 females; the group without: 43 males vs.77 females). Some parameters observed before, such as dyslipidemia and obesity, are still most prevalent in people who adhere to the MD requirements. Only reduced uric acid concentration, body mass, and WC were presented in the group with adherence to MD. Groups with MD adherence were also characterized by fewer patients with TG≥150 mg/ml.
Having 207 women is better to examine only this group composed of one gender. The second solution is to properly compos the tested group with the correct proportion of females and males.
Author Response
Reviewer #2:

Round 2
Reviewer 1 Report
Comments and Suggestions for Authors
Thak you for improving the quality of the manuscript.
Reviewer 2 Report
Comments and Suggestions for Authors The paper do not meet Nutrient’s requirements.